# Daily Green Tea Infusions in Hypercalciuric Renal Stone Patients: No Evidence for Increased Stone Risk Factors or Oxalate-Dependent Stones

**DOI:** 10.3390/nu11020256

**Published:** 2019-01-24

**Authors:** Julie Rode, Dominique Bazin, Arnaud Dessombz, Yahia Benzerara, Emmanuel Letavernier, Nahid Tabibzadeh, Andras Hoznek, Mohamed Tligui, Olivier Traxer, Michel Daudon, Jean-Philippe Haymann

**Affiliations:** 1Sorbonne Université, Service d’Explorations Fonctionnelles Multidisciplinaires, AP-HP, Hôpital Tenon, 75020 Paris, France; laurent.benzerara@aphp.fr (Y.B.); emmanuel.letavernier@aphp.fr (E.L.); nahid.tabibzadeh@inserm.fr (N.T.); michel.daudon@aphp.fr (M.D.); jean-philippe.haymann@aphp.fr (J.-P.H.); 2Laboratoire de Chimie Physique, Université Paris-Sud, Bat 349, 91405 Orsay, France; dominique.bazin@u-psud.fr; 3Centre National de la Recherche Scientifique (CNRS), Département de Physique, 91405 Orsay, France; arnaud.dessombz@gmail.com; 4Sorbonne Université, GRC n°20, Groupe de Recherche Clinique sur la Lithiase Urinaire, Hôpital Tenon, F-75020 Paris, France; olivier.traxer@aphp.fr; 5Institut National de la Santé et de la Recherche Médicale (INSERM), UMR-S 1155 Paris, France; 6Service d’Urologie, Hôpital Henri Mondor, Assistance Publique-Hôpitaux de Paris, 94000 Créteil, France; andras.hoznek@aphp.fr; 7Sorbonne Université, Service d’Urologie, AP HP, Hôpital Tenon, 75020 Paris, France; mohamed.tligui@aphp.fr

**Keywords:** green tea, oxalate, renal stone, calcium oxalate monohydrate, hypercalciuria

## Abstract

Green tea is widely used as a ‘’healthy’’ beverage due to its high level of antioxidant polyphenol compounds. However tea is also known to contain significant amount of oxalate. The objective was to determine, in a cross-sectional observational study among a population of 273 hypercalciuric stone-formers referred to our center for metabolic evaluation, whether daily green tea drinkers (*n* = 41) experienced increased stone risk factors (especially for oxalate) compared to non-drinkers. Stone risk factors and stone composition were analyzed according to green tea status and sex. In 24-h urine collection, the comparison between green tea drinkers and non-drinkers showed no difference for stone risk factors such as urine oxalate, calcium, urate, citrate, and pH. In females, the prevalence of calcium oxalate dihydrate (COD) and calcium phosphate stones, assessed by infrared analysis (IRS) was similar between green tea drinkers and non-drinkers, whereas prevalence of calcium oxalate monohydrate (COM) stones was strikingly decreased in green tea drinkers (0% vs. 42%, *p* = 0.04), with data in accordance with a decreased oxalate supersaturation index. In males, stone composition and supersaturation indexes were similar between the two groups. Our data show no evidence for increased stone risk factors or oxalate-dependent stones in daily green tea drinkers.

## 1. Introduction

The high prevalence of urolithiasis (reaching up to 8–10% of the general population) is mainly related to environmental factors, especially the Western diet [1]. Calcium stones are encountered in 80% of cases and often contain a mixture of calcium oxalates and calcium phosphates. Among calcium oxalate crystals, the calcium oxalate monohydrate crystalline form (COM) is oxalate-dependent, whereas the calcium oxalate dihydrate crystalline form (COD) is calcium-dependent [1]. Hence, high urinary calcium and oxalate concentrations are critical factors leading to stone formation. Tea contains oxalates in varying amounts depending on the type and duration of the infusion. The amount of oxalate measured for black tea varies from 2.7 to 4.8 mg/240 mL (one cup) of tea infused for 1–5 min [2], whereas the amount of oxalate in green tea ranges from 2.08 to 34.94 mg/250 mL of tea [3]. However, the amount of oxalate in green tea depends on its origin, quality, preparation, and time of harvest, thus probably explaining why some studies report a higher oxalate concentration in black tea compared to green tea [4]. However, tea extracts, particularly green tea, are considered to have many beneficial clinical effects for centuries. Tea infusions contain polyphenol compounds, among which catechins have been of major interest due to their antioxidant properties [5]. Indeed, among the different varieties of teas, green teas as compared to black teas contain the highest concentration of catechins [6]. Other food and drinks (in particular wine and dark chocolate) may also represent a substantial dietary source of catechins, but to a lesser extent [7,8]. Catechins are mainly found under four different hydro soluble forms in green tea: epigallocatechin-3-*O*-gallate (EGCg), epicatechin-3-*O*-gallate (ECG), epigallocatechin (EGC), and epicatechin (EC) [9,10]. EGC and its metabolites are the main compounds found in the urine following the ingestion in humans and animals with concentrations up to 100 µmoles/L in humans [11,12,13,14,15,16,17,18,19,20]. As a matter of fact, several authors recommend green tea or large amounts of catechin intake in order to prevent crystallization of calcium oxalate crystals in animal models [21,22,23]. Conversely, tea is also known to contain high amounts of oxalate, and could have diuretic effects increasing natriuresis but potentially also calciuria, and thus its consumption is regarded by other authors as a genuine risk factor for renal stone formation [21,24,25]. The aim of this work was to study the influence of regular daily green tea intake on stone risk factors, stone morphology, and composition and to assess a potential increased risk for oxalate-dependent stones.

## 2. Material and Methods

### 2.1. Population

The data of 420 hypercalciuric renal stone formers referred to our department between 2009 and 2011 for a routine metabolic evaluation (including an oral calcium load test) were retrospectively analyzed. A careful clinical examination (including a survey related to diet and fluid intake, and noteworthy daily green tea intake) was performed. All patients gave their informed written consent for inclusion before they participated in the study. This observational cross sectional study was conducted in accordance with the Declaration of Helsinki and French legislation.

In total, 273 patients (flow chart Figure 1) were included after the exclusion of patients with a diagnosis of primary hyperparathyroidism, sarcoidosis, bowel resection, on-going steroid, bisphosphonate, antiviral, diuretic, and/or vitamin D treatments, vegetarian diet, consumption of “exotic” infusions or food supplements (in tablets or powder), and/or intermittent green tea intake or daily black tea intake. The green tea group (*n* = 41) was defined as patients drinking at least one cup (250–300 mL) of green tea daily, and non-drinkers as patients drinking no green tea at all (*n* = 232). Renal stone composition was available for 98 out of 273 patients (36%). A comparison between the two groups was performed according to sex (Figure 1).

A morpho-constitutional analysis was performed for each stones as described previously [26]. In short, the standardized protocol comprises two steps. First, a morphologic examination by means of a stereomicroscope (magnification × 10–40) of the surface and section of the calculus, with the identification of the nucleus (or core) and description of the inner organization, was carried out. The main points to be recorded in each stone were size, form, color, aspect (smooth, rough, or spiky) of the surface, presence of a papillary imprint (umbilication), presence of Randall’s plaque, aspect of the section (well organized with concentric layers and/or radiating organization, or poorly organized and loose structure), and location and aspect of the nucleus. Thereafter, an analysis was performed by infrared spectroscopy (IRS) of a sample of each part of the calculus and in particular the global proportion of components in a powdered sample of the whole stone.

All of the urine collections were performed at least 3 months after lithotripsy or surgery. A 24-h urine collection under a regular diet was performed at baseline to measure the following parameters: diuresis volume, calcium, magnesium, phosphate, sodium, potassium, creatinine, urea, oxalate, uric acid, citrate, ammonium, and deoxypyridinoline excretion. A fasting blood sample was analyzed for total and ionized calcium, phosphate, magnesium, creatinine, uric acid, bicarbonates, parathyroid hormone (PTH), 25(OH)-D3, and 1,25(OH)-D3 vitamins. Bone remodeling biomarkers (serum bone alkaline phosphatase (BALP)) were also performed at that time. 

Serum and urinary creatinine levels were measured by enzymatic method on a Konelab 20 analyzer from Thermo Fisher Scientific (Vantaa, Finland). Uric acid levels were measured with the Konelab analyzer (Thermo Fisher Scientific, Vantaa, Finland). Total CO_2_ in blood, ionized calcium, sodium, and potassium levels were measured with an ABL 815 from Radiometer (Bronshoj, Denmark). Calcium and magnesium serum and urinary levels were measured with the PerkinElmer 3300 atomic absorption spectrometer (Courtabeuf, France). In addition, 25(OH)-D3 and 1,25(OH)-D3 were measured by radioimmunoassay kits from Immunodiagnostics Systems Ltd. (Paris, France). Parathyroid hormone was measured by the ELSA-PTH kit from Cisbio International (Codolet, France). Urinary NH4 was measured with the RANDOX Laboratories kit (Crumlin, UK). Urinary deoxypyridoline was measured by the RIA method from Immunodiagnostics Systems Ltd. (Paris, France). BALP level was measured with Ostase bone alkaline phosphatase enzyme immunoassay obtained from Immunodiagnostics Systems Ltd. (Paris, France). Citrate and oxalate measurements were performed by ionic chromatography (Metrohm, Courtabeuf, France). Ionic strength and supersaturation indexes for calcium oxalate, urate, and brushite were calculated using molar concentrations [24,25].

### 2.2. Statistical Analyses

Statistical analyses were performed by two different operators using StatView (SAS Institute, Inc., Cary, NC, USA) and *R* software (The *R* Foundation, Lincoln, NE, USA). Quantitative data were expressed as the mean and SD unless otherwise indicated and as a percentage for categorical variables. Because of sex differences for many biologic parameters, analyses were performed separately in women and men. Comparisons were performed using the *t*-test or a nonparametric Wilcoxon and Mann–Whitney test, whenever required. Comparisons of qualitative parameters were performed using a chi-squared test or Fisher’s exact test when necessary. *p* < 0.05 was considered statistically significant.

## 3. Results

### 3.1. Demographic and Clinical Data

Among our population, 13.5% of males and 17% of females were regular green tea drinkers (i.e., drinking at least one cup a day) (*p* = 0.61). Median age in hypercalciuric renal stone patients was 47 years old (ranging from 18 to 82 years). Cardiovascular risk factors were not infrequent findings in this population: overweight status was present in 54.4% of our population, dyslipidemia was encountered in 24.4% of cases, ongoing smoking or tobacco exposure in 24% and 41% of cases, respectively, 5.7% had type 2 diabetes, and 24% had high blood pressure. However, no difference was detected between green tea drinkers and non-drinkers (Table 1).

### 3.2. Diet, Metabolic, and Urinary Stone Risk Factors

Comparison between male and female groups showed similar urine output (1.9 vs. 1.8 L/day, *p* = 0.22), urinary calcium (6.4 vs. 6.7 mmol/day, *p* < 0.45), oxalate (0.33 vs. 0.33 mmol/day, *p* = 0.63), urate (3.8 vs. 3.6 mmol/day, *p* < 0.31), urea (404 vs. 400 mmol/day, *p* < 0.86), and sodium (134 vs. 140 mmol/day, *p* < 0.44). Surprisingly, a higher fluid intake (declarative survey) in female green tea drinkers compared to non-drinkers was not confirmed by a higher daily urine output (Table 2). Nevertheless, as shown Table 2, the analysis according to sex showed no difference for stone risk factors between green tea drinkers and non-drinkers such as oxalate, calcium, urate, and citrate with even a trend for a lesser oxaluria in the green tea female population (0.32 vs. 0.27, *p* = 0.09). Moreover, 24-h urine supersaturation indexes were similar between green tea drinkers and non-drinkers noteworthy for the calcium oxalate relative supersaturation index (CaOx RSS) and the calcium oxalate (CaOx) product (Table 3). However, a significant higher Ca/Ox ratio is noticed in female green tea drinkers, suggesting a relatively lower risk for oxalate-dependent stones (though the other Ca Ox indexes appear similar).

As shown Table 2, other biological data were similar between drinkers and non-drinkers in the male and female population in terms of noteworthy renal function, calcium phosphate homeostasis, and bone remodeling biomarkers.

Among the 98 renal stones available, 48 samples were collected from female and 50 from male patients. Within male or female groups, comparison of the major stone component identified by IRS analysis revealed no significant difference between drinkers and non-drinkers. Of note, no COM stones were detected in the female drinkers group compared to female non-drinkers (0% vs. 42%, *p* = 0.04) whereas in male drinkers, the prevalence of COM was similar between groups (33% vs. 44%, *p* = NS) (Table 4).

## 4. Discussion

The aim of the present study was to acknowledge whether drinking green tea on a daily basis would exert any influence on stone risk factors and/or calcium stone structure or composition. In the first part of our work, based upon a cross sectional observational study, we found no difference between green tea drinkers and non-drinkers for stone risk factors in particular oxalate excretion in 24-h urine collections but also urine pH, calcium, urate, and citrate. These results were further confirmed by supersaturation indexes. These data have a clinical relevance as renal stone patients are commonly advocated against regular tea drinking based upon the oxalate content reported in tea leaves [3,4]. According to the view that green tea intake would increase stone activity, one study reported an increased urinary calcium excretion in an experimental setting [21]. Conversely, in two other animal studies, the administration of catechins or green tea prevented crystallization, especially monohydrate CaOx crystal deposits within tubular lumen [22,23]. However a recent study in a very large prospective Chinese cohort reports that green tea intake was associated with a lower risk of incident kidney stones [27]. Our results are in accordance with these findings and suggest that drinking daily green tea (assessed by a detailed survey) would not be detrimental in both sexes. As a matter of fact, drinking daily green tea was reported to have also other pharmacological effects such as weight loss, cardiovascular protection, and bone mineralization [28,29]. Though this study is not designed to assess these issues, our data show no difference for body mass index, cardiovascular risk factors, bone remodeling biomarkers, or calcium and phosphate blood levels between drinkers and non-drinkers. 

The second part of our work was to study a potential calcium stones composition difference between green tea drinkers and non-drinkers. Of note, among 98 stones available from this idiopathic hypercalciuric population, 34% contained COM as a major component. However, the major component of COM was similar between regular green tea drinkers and non-drinkers in the whole population (*p* = 0.26). Accordingly, similar CaOx supersaturation indexes are detected, thus ruling out green tea as a potential additional stone risk factor for COM stones. Surprisingly, in female green tea drinkers, no COM stone was detected at all (Table 4), thus suggesting either a pharmacological effect illustrated by the increased Ca/Ox ratio (Table 3) and/or a potential role of green tea catechins (or antioxidants) directing CaOx crystallization from COM to COD as previously shown in vitro [30]. Alternatively, catechins could exert a potential inhibition of COD to COM conversion. This exciting speculation however requires to be specifically addressed in further studies.

Indeed, a high prevalence of COD in the female green tea group is very unusual as COM and calcium phosphate stones are the usual major compounds reported in the female renal stone population [26,31]. Conversely, in male stone-formers, COD and COM are the two main compounds encountered in both groups, with a similar prevalence between the two groups. Thus according to stone composition drinking daily green tea has no demonstrated over risk for oxalate-dependent stones in our studied population. 

Our study suffers some limitations as it is an observational study with a declarative diet survey and thus did not take into account the total amount of catechins intake in the diet. Indeed, the amount of catechins in green tea beverage depends upon green tea leaves, temperature, and the duration of infusion [3]. Moreover, substantial amount of catechins are found in a significant number of food including wine, which consumption is usually underestimated, and thus represent a bias in our study. However, despite the lack of dietary questionnaire (except for calcium intake and water), sodium and protein intake appeared similar between the two groups as assessed by 24-h urine sodium and urea. Last, similar oxaluria values in 24-h urine collection support the view that regular green tea intake is not a risk factor for oxalate-dependent stones (assessed also by stone composition). The 50% prevalence of COD stones in female green tea drinkers is related to idiopathic hypercalciuria; however, this finding may also raise the issue as to whether green tea would be an additional risk factor for an increased prevalence of calcium-dependent stones (illustrated by an increased Ca/Ox ratio in the female green tea group). Further studies are required to assess whether in non-hypercalciuric renal stone patients and/or in the general population green tea would prevent COM stone occurrence or recurrence. This specific issue is however beyond the goal of the present study. 

## 5. Conclusions

Our data show no evidence for increased oxalate-dependent stones in daily green tea drinkers, with no increased oxaluria or calciuria in 24-h urine collection and, to our surprise, no reported COM stones in our female green tea drinkers group. A clinical trial testing the hypothesis that high catechin intakes may prevent COM stone recurrence would be most welcome.

## Figures and Tables

**Figure 1 nutrients-11-00256-f001:**
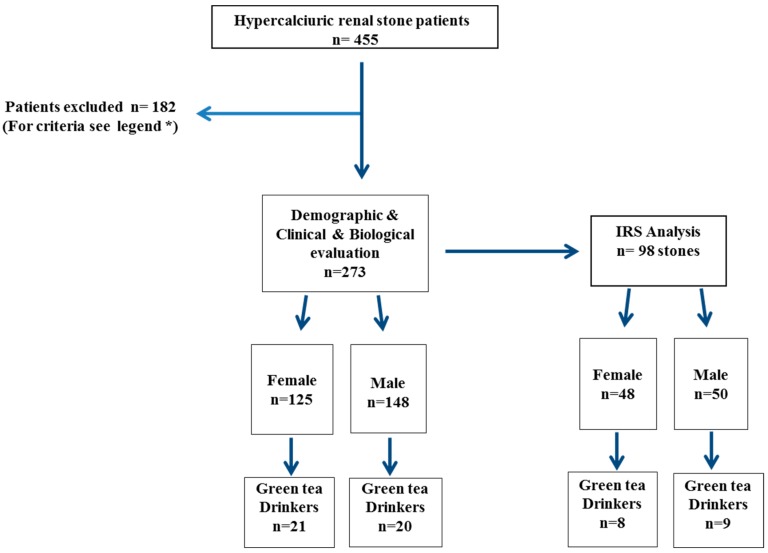
Flow chart. Primary hyperparathyroidism or sarcoidosis (*n* = 28); bowel resection (*n* = 2), steroid, bisphosphonate, and antiretroviral treatment (*n* = 23); vegetarian diet or “exotic” infusions (*n* = 25); food supplements (*n* = 22); intermittent green tea intake (*n* = 6); daily black tea intake (*n* = 36); diuretics, vitamin D treatment (*n* = 39); pregnancy (*n* = 1). IRS: infrared analysis.

**Table 1 nutrients-11-00256-t001:** Demographic and clinical data.

>Population	Female (*n* = 125)			Male (*n* = 148)		
	Non-Drinkers (*n* = 102)	Green Tea (*n* = 21)	*p*	Non-Drinkers (*n* = 122)	Green Tea (*n* = 20)	*p*
Age (years)	46 (34–59)	42 (33–53)	NS	48 (38–58)	44 (37–57)	NS
BMI (kg/m^2^)	24.9 (21.4–29.4)	23.5 (21.2–25.6)	NS	25.7 (23.2–29.0)	25.8 (24.5–29.6)	NS
MAP (mmHg)	83.3 (76.7–92.5)	81.7 (76.7–93.4)	NS	90.0 (80.0–96.7)	86.7 (82.5–93.3)	NS
Hypertension	37%	33%	NS	47%	50%	NS
Dyslipidemia	22%	9%	NS	26%	15%	NS
Diabetes	4%	9%	NS	8%	0%	NS
Age first stone (years)	27.0 (19.0–41.0)	30.0 (18.7–41.2)	NS	29.5 (21.0–41.5)	27.5 (22.5–39.5)	NS
SWL (% of patients)	37.3%	33.3%	NS	47.5%	50.0%	NS
URS (% of patients)	45.1%	47.6%	NS	51.6%	40.0%	NS

BMI: body mass index. MAP: mean arterial pressure. NS: not significant. SWL: Shock waves lithotripsy. URS: flexible ureteroscopy.

**Table 2 nutrients-11-00256-t002:** Food intake evaluation, biological data, and metabolic risk factors.

Population	Female (*n* = 125)			Male (*n* = 148)		
	Non-Drinkers (*n* = 102)	Green Tea (*n* = 21)	*p*	Non-Drinkers (*n* = 122)	Green Tea (*n* = 20)	*p*
Fluid intake ≥ 2 L/day (%)	11.2%	50.0%	<0.0001	30.3%	22.2%	0.47
Blood						
Sodium (mmol/L)	139 (138–140)	139 (138–140)	0.84	139 (138–140)	140 (138–140)	0.35
Potassium (mmol/L)	4.0 (3.8–4.3)	4.1 (3.8–4.3)	0.37	4.0 (3.9–4.3)	4.1 (3.9–4.3)	0.39
Fasting glucose (mmol/L)	5.5 (5.1–6.3)	5.1 (5.06–5.5)	0.06	5.4 (5.0–5.9)	5.8 (5.5–6.4)	0.008
tCO_2_ (mmol/L)	27.5 (26.0–29.0)	27.9 (25.1–29.7)	0.70	27.6 (26.3–29.7)	27.6 (25.3–29.6)	0.78
Creatinine clearance (mL/min)	126 (95–145)	109 (84–138)	0.16	121 (97–153)	118 (101–157)	0.94
Ionized calcium (mmol/L)	1.18 (1.15–1.21)	1.19 (1.16–1.21)	0.43	1.18 (1.15–1.21)	1.17 (1.14–1.21)	0.82
PTH (pg/mL)	35 (26–49)	31 (25–40)	0.31	35 (27–49)	40 (33–55)	0.11
25 OH vitamin D (pg/mL)	24 (17–34)	25 (19–37)	0.52	25 (16–37)	24 (20–30)	0.95
1-25 (OH)_2_ vitamin D (ng/mL)	66 (55–85)	79 (54–90)	0.5	66 (52–84)	58 (57–87)	0.48
BALP (UI/L)	13.5 (10.3–17.0)	12.3 (10.3–15.1)	0.36	13.5 (10.0–17.0)	10.6 (9.5–13.0)	0.06
Deoxypyridin (mmol/mmol creat)	5.7 (4.5–8.1)	6.6 (4.1–9.7)	0.7	5.4 (4.2–7.5)	5.2 (3.9–6.4)	0.29
Urine						
Diuresis (mL/day)	1880 (1460–2582)	1908 (1757–2368)	0.89	1865 (1433–2337)	1836 (1231–2301)	0.62
Calcium (mmol/day)	6.2 (4.6–8.1)	7.0 (5.5–10.0)	0.08	5.7 (4.0–8.5)	6.3 (4.5–8.1)	0.95
Oxalate (mmol/day)	0.32 (0.24–0.42)	0.27 (0.32–0.34)	0.09	0.30 (0.19–0.41)	0.29 (0.21–0.43)	0.77
Urate (mmol/day)	3.5 (2.6–4.7)	3.2 (2.7–3.7)	0.33	3.5 (2.9–4.5)	3.8 (3.0–4.7)	0.51
Citrate (mmol/day)	2.4 (1.1–3.4)	2.0 (1.7–3.4)	0.98	2.5 (1.5–3.5)	2.2 (0.4–2.7)	0.22
Fasting pH	6.33 (5.68–6.66)	6.2 (6.0–6.6)	0.83	6.18 (5.62–6.61)	5.74 (5.32–6.32)	0.06
Sodium (mmol/day)	113 (84–157)	121 (84–149)	0.25	127 (95–173)	126 (90–146)	0.4
Ammonium (mmol/day)	35 (26–45)	30 (26–45)	0.28	35 (25–48)	44 (31–54)	0.39
Magnesium (mmol/day)	4.1 (3.2–5.0)	5.0 (3.3–7.0)	0.08	4.4 (3.2–5.9)	4.2 (2.7–4.9)	0.48

BALP: bone alkaline phosphatase. tCO_2_: plasma bicarbonate. PTH: parathyroid hormone.

**Table 3 nutrients-11-00256-t003:** Twenty-four hour urine supersaturation indexes.

Population	Female			Male		
	Non-Drinkers	Green Tea	*p*-Value	Non-Drinkers	Green Tea	*p* Value
**AP CaOx index**	0.74 (0.39–1.30)	0.71 (0.28–1.31)	0.42	0.67 (0.37–1.10)	0.76 (0.41–1.28)	0.73
**Br RSS**	1.3 (0.3–2.5)	1.0 (0.1–1.7)	0.27	0.9 (0.3–1.9)	0.6 (0.2–1.3)	0.39
**UA RSS**	0.54 (0.25–1.81)	0.75 (0.19–1.48)	0.95	1.08 (0.41–2.26)	1.99 (0.33–3.96)	0.47
**CaOx RSS**	5.6 (3.7–8.6)	3.7 (2.0–8.3)	0.16	5.3(3.2–7.7)	6.0 (3.7–8.7)	0.64
**Ca.Ox**	0.53 (0.26–1.03)	0.59 (0.23–1.15)	0.67	0.47 (0.25–0.89)	0.57 (0.30–0.91)	0.57
**Ratio Ca/Ox**	19 (12.5–30)	26 (20.5–40.5)	0.01	19 (12–34)	18.5 (12–28.5)	0.65
**Ionic Strength**	0.08 (0.06–0.12)	0.09 (0.07–0.12)	0.72	0.09 (0.05–0.12)	0.07 (0.075–0.105)	0.43

RSS: relative super saturation; AP CaOx index: Tiselius index; Br RSS: brushite relative super saturation; CaOx RSS: calcium oxalate relative super saturation; Ca.Ox: calcium oxalate product. Ca/Ox: calcium/oxalate ratio. UA RSS: uric acid relative super saturation.

**Table 4 nutrients-11-00256-t004:** IRS analysis of stones collected from drinkers and non-drinkers.

Population	Female			Male		
	Non-Drinkers (*n* = 40)	Green Tea (*n* = 8)	*p* Value	Non-Drinkers (*n* = 41)	Green Tea (*n* = 9)	*p* Value
Major COM component (%)	42	0	0.036	33	44	0.99
Major COD component (%)	26	50	0.23	41	55	0.72
Carbapatite major component (%)	16	12.5	0.99	7	0	0.99
Type Ia or Ib (%)	23	0	0.32	18	11	0.99
Type IIa or IIb (%)	50	62	0.7	64	78	0.69
Type IVa (%)	10	12	0.99	8	0	0.99

Type Ia, Ib morphology refers to COM subtype crystalline forms. Type IIa and IIb refer to COD subtype crystalline forms (Type IIb also includes the presence of COM crystalline form). Type IVa refers to carbapatite subtype crystalline forms.

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
