# Peer review of "Daily Green Tea Infusions in Hypercalciuric Renal Stone Patients: No Evidence for Increased Stone Risk Factors or Oxalate-Dependent Stones"

_nutrients, 2019, doi:10.3390/nu11020256_

Reviewer 1 Report

The paper is interesting because it studies an aspect that has been considered true without clear scientific evidence: the association between the consumption of tea and the increase in oxalocalcic lithiasis. This study shows that the consumption of green tea, in the amounts that have been established, does not increase the excretion of oxalate. Therefore, in this situation the consumption of tea does not seem to promote the development of  calcium oxalate kidney stones.

The main concern is related to the relationship between the type of renal calculi and the consumption of tea. Thus, the number of calculi, considering the subdivision into groups, is very small and, if the study has been carried out exclusively by IRS analysis, it is possible that some calculi of COM come from the transformation of COD. In fact, Table 4 specifies the characteristics of the nuclei, as well as the subtypes observed. It follows that the study of the stones does not correspond to a simple IRS analysis of the calculi. However, the Material and Methods Section does not specify how the stones have been studied, and should be clearly indicated. Nor is it indicated how 24-hour urine has been analyzed and should be included.

Author Response

Reviewer 1

The paper is interesting because it studies an aspect that has been considered true without clear scientific evidence: the association between the consumption of tea and the increase in oxalocalcic lithiasis. This study shows that the consumption of green tea, in the amounts that have been established, does not increase the excretion of oxalate. Therefore, in this situation the consumption of tea does not seem to promote the development of  calcium oxalate kidney stones.

The main concern is related to the relationship between the type of renal calculi and the consumption of tea. Thus, the number of calculi, considering the subdivision into groups, is very small and, if the study has been carried out exclusively by IRS analysis, it is possible that some calculi of COM come from the transformation of COD. In fact, Table 4 specifies the characteristics of the nuclei, as well as the subtypes observed. It follows that the study of the stones does not correspond to a simple IRS analysis of the calculi. However, the Material and Methods Section does not specify how the stones have been studied, and should be clearly indicated. Nor is it indicated how 24-hour urine has been analyzed and should be included.

As suggested we added in the material and Methods section the procedure for morpho constitutional stone analysis and also the method for biological parameters determination.

We indeed agree that some calculi of COM could come from the transformation of COD. Green tea, however, is probably influencing crystalline conversion of COD into COM: the finding that % of COM major component was higher in non drinkers compared to green tea drinkers (42 versus 0%, p=0.03) raises the interesting issue of a potential inhibition COD conversion into COM by green tea. Indeed, type Ia and b versus IIa and b morphology in female non drinkers was 23% and 50% respectively (table 4) whereas in green tea drinkers the figures were 0% and 62% respectively. These figures suggest that no COD into COM conversion had occurred in green tea drinkers whereas 27% (50%-23%) of COD to COM conversion may have occurred in this latter group. We added this speculation with caution in our revised discussion and do thank you for your relevant comments.

Reviewer 2 Report

This paper is a cross-sectional analysis that investiaged the relationship between green tea drinkers among stone formers. The research design is very unclear for observational research. It is a cross-sectional analysis among cases (stone formers), but the authors talk about controls, which would indicate a case-control design. Furhter the author use the word prospective which would imply a cohort analysis. Please, consult an experienced epidemiologist to help with the set-up of the design and the analyses.

Abstract: mention precize study design not observational but more specific

Introduction:

Please, give more background for oxalate products/foods. give examples and how much is in 1 cup of tea, etc.

Methods

The determinant and outcomes should be described in more detail to get sufficient information about the measurements.

How was green tea intake defined. With a validated method. Was the amount of green tea taken into account. What has been done with other tea-like bevarages?

Stones

Please, give more information on the location of the stones.

The initial recruitment included 420 patients while the analytic sample contained 273 patients. Present potential differences between those included and excluded.

What is the rationale for stratified analyis by sex? Please use the word sex instead of gender since only sex has been assessed. Was there significant interaction?

Exclusions:

The authors exclude so many patients without an explanation. Why are black tea drinkers and diuretic use excluded. Please, perform sensistivity analyses to test the robustness of the association rather than excluding those patients. Further, black tea intake could be a confounder and should be adjusted for.

Statistical analyses.

This section should describe every step that has been done. Please consult a biostatistician or epidemiologist for support. The results would benefit from adjusted analyses which can be done with regression or ANCOVA.

Tables: please, introduce all abbreviations. Delete P-values for table 1 since this is absolutely not necessary.

Who are controls? Patients without stones?

Table 3:

So many outcomes have been presented. What are the main outcomes. What is most important?

These analyses should be presented adjusted for confounders including: age, sex, black tea, red wine etc.

Discussion

Should be revised according to the comments above and should include more in-depth information about the potential mechanisms.

Author Response

Please find reply to reviewer in attached file

Round  2

Reviewer 1 Report

The paper has been significantly improved.